# Anti-Fibrotic Effect of Synthetic Noncoding Decoy ODNs for TFEB in an Animal Model of Chronic Kidney Disease

**DOI:** 10.3390/ijms23158138

**Published:** 2022-07-23

**Authors:** Sun-Jae Lee, Young-Ah Kim, Kwan-Kyu Park

**Affiliations:** Department of Pathology, School of Medicine, Daegu Catholic University, Daegu 42472, Korea; pathosjlee@cu.ac.kr (S.-J.L.); youngah7840@naver.com (Y.-A.K.)

**Keywords:** TFEB, UUO, autophagy, renal fibrosis, ODNs

## Abstract

Despite emerging evidence suggesting that autophagy occurs during renal interstitial fibrosis, the role of autophagy activation in fibrosis and the mechanism by which autophagy influences fibrosis remain controversial. Transcription factor EB (TFEB) is a master regulator of autophagy-related gene transcription, lysosomal biogenesis, and autophagosome formation. In this study, we examined the preventive effects of TFEB suppression on renal fibrosis. We injected synthesized TFEB decoy oligonucleotides (ODNs) into the tail veins of unilateral ureteral obstruction (UUO) mice to explore the regulation of autophagy in UUO-induced renal fibrosis. The expression of interleukin (IL)-1β, tumor necrosis factor-α (TNF-α), and collagen was decreased by TFEB decoy ODN. Additionally, TEFB ODN administration inhibited the expression of microtubule-associated protein light chain 3 (LC3), Beclin1, and hypoxia-inducible factor-1α (HIF-1α). We confirmed that TFEB decoy ODN inhibited fibrosis and autophagy in a UUO mouse model. The TFEB decoy ODNs also showed anti-inflammatory effects. Collectively, these results suggest that TFEB may be involved in the regulation of autophagy and fibrosis and that regulating TFEB activity may be a promising therapeutic strategy against kidney diseases.

## 1. Introduction

Chronic kidney disease (CKD), characterized by glomerular and tubulointerstitial fibrosis (TIF), is a global health problem with a prevalence of 5–10% [1]. The deposition of extracellular matrix proteins is a hallmark of CKD, regardless of the etiology of the primary disease [2,3,4]. Fibroblast proliferation is a precursor to extracellular matrix (ECM) overproduction [5]. The underlying cellular events are complex and involve the interaction of multiple renal resident cells with infiltrating inflammatory cells as well as tubular epithelial-to-mesenchymal transition (EMT), monocyte/macrophage and T cell infiltration and cellular apoptosis, and interstitial fibroblast and glomerular mesangial cell differentiation into activated myofibroblasts [6]. Apart from EMT, several studies have demonstrated that endothelial-to-mesenchymal transition (EndoMT) also plays an important role in the recruitment of fibroblasts [7,8]. However, the mechanisms regulating EndoMT have not been fully elucidated, and only a few studies have reported molecular changes and regulatory events occurring in endothelial cells during phenotypic transformation into activated myofibroblasts [9]. In addition, several inflammatory and metabolic molecular mechanisms, including TGF-β [10], WNT signaling [11], fibroblast growth factor receptor signaling [12], Notch signaling [13], Hedgehog signaling [14], endothelial [15] and podocyte glucocorticoid receptors [16], endothelial sirtuin 3 (SIRT3)-mediated mechanisms [17] and DPP-4 mediated mechanism [18], are known to be implicated in the pathogenesis of diabetic nephropathy. Factors that promote/inhibit fibrosis restrict each other, and the dynamic balance is lost, resulting in the formation of fibrosis [19,20]. The injured site of the renal interstitium can be rapidly infiltrated by a large number of inflammatory cells, aggravating fibrosis [19]. After prolonged renal injury, the infiltrated inflammatory cells release excessive pro-inflammatory cytokines, such as tumor necrosis factor (TNF) -α, interleukin (IL)-1β and IL-6, to clean up tissue debris, dead cells, and invading organisms from the injured site, as well as pro-fibrotic cytokines and growth factors [6,21,22,23]. Once renal fibrosis develops, most patients progress to irreversible end-stage renal disease, in which kidney transplantation with dialysis is the only therapeutic option [22]. Therefore, slowing down the progression of fibrosis could be an important strategy for preventing CKD [24].

Angiotensin-converting enzyme inhibitors (ACEIs) and angiotensin receptor blockers (ARBs) are the traditional therapeutic agents for CKD patients. ACEIs and ARBs in CKD reduce systemic and intra-glomerular blood pressure and proteinuria [25]. However, treatment with ACEIs or ARBs does not prevent direct renal injuries or metabolic changes associated with diabetic nephropathy such as an over-activation of the mineralocorticoid receptor [26,27]. To overcome these limitations, several potential drugs have been evaluated, including SIRT3 [28] and glycolysis inhibitors [29,30], Linagliptin [31], ROCK inhibitors [32,33], mineralocorticoid antagonists [26], and peptide AcSDKP [18,27], for protecting against renal injuries. However, further studies are still necessary because most of these agents are still in preclinical status and the mechanisms of renal fibrosis are not fully understood.

In recent years, the relationship between autophagy and renal fibrosis has been reported in studies on renal inflammation and autophagy [22,24,34]. Autophagy is an important cellular mechanism for the intracellular lysosome-mediated degradation of damaged organelles, protein aggregates, and other macromolecules in the cytoplasm. It regulates cell death under normal physiological and pathological conditions [35,36]. Lysosomes fuse with phagosomes to degrade their contents, contributing to lysosomal membrane permeabilization [37,38]. Autophagy is involved in renal diseases including acute kidney injury, glomerular diseases, and TIF [22]. Despite emerging evidence suggesting that autophagy occurs during renal TIF, the role of autophagy activation in fibrosis and the mechanism by which autophagy influences fibrosis remain controversial.

Transcription factor EB (TFEB) is a master regulator of autophagy-related gene transcription, lysosome biogenesis, and autophagosome formations [39,40,41,42]. TFEB levels increase during inflammation and fibrosis [34,43]. Previous studies have reported increased autophagic activity in tubular cells in human biopsy samples from transplanted kidneys [44] and patients with renal disease [45]. These findings suggest the role of the TFEB-autophagy signaling pathway in the pathogenesis of renal tubular injuries and CKD in humans. However, the pathophysiological roles of TFEB in modulating autophagy and tubulointerstitial injury in CKD remain controversial, and whether autophagy inhibition has a therapeutic effect on renal injury is unclear. Thus, it is necessary to examine the effect of autophagy inhibition and its underlying mechanism in an animal model of renal injury.

Synthesized TFEB decoy oligonucleotides (ODNs) block the transcription factors of a specific gene that can recognize their consensus-binding sequences. Previous studies [46,47,48] have demonstrated that decoy ODNs substantially down-regulate the functions of transcription factors in several disorders. Lee et al. [49] reported the efficacy of synthetic decoy ODNs in an animal atherosclerosis model. Additionally, the unilateral ureteral obstruction (UUO) mouse model is a representative animal model of obstructive nephropathy and is characterized by progressive TIF [50]. In the kidneys of UUO mice, autophagy is accompanied by increased renal tubular injury and fibrosis [51]. Therefore, UUO is a suitable model that is believed to mimic human chronic obstructive nephropathy.

This study aimed to investigate the association between kidney function and autophagy using synthetic TFEB decoy ODNs, which were designed to inhibit TFEB transcription factors in UUO kidneys, to determine the role of TFEB-mediated autophagy in CKD-related fibrosis and its underlying mechanism.

## 2. Results

### 2.1. TFEB Decoy ODN Attenuats Morphological Changes in UUO-induced Renal Fibrosis

Thirty-five mice were randomly divided into five groups: the NC group, the TFEB group that was injected with TFEB decoy ODNs, the UUO group that underwent a UUO surgery, the Scr group that included mice that underwent UUO surgery and were injected with Scr ODNs, and the UUO + TFEB group comprising mice that underwent UUO surgery and were injected with TFEB decoy ODNs.

We evaluated the histologic effect of TFEB decoy ODNs on renal TIF in UUO mouse kidneys using H&E (Figure 1A) and Masson’s trichrome staining (Figure 1B). The NC and TFEB groups showed unremarkable histologic changes. However, features of severe tubular interstitial injury, including tubular atrophy and TIF, were observed in the UUO and Scr groups. Compared to the UUO group, the UUO + TFEB group showed reduced histologic change in renal damage. Masson’s trichrome staining showed increased collagen accumulation in the UUO and Scr groups, whereas the expression of collagen was clearly decreased in the UUO + TFEB group (Figure 1B). Western blot analysis revealed increased expression of fibronectin and collagen I in the UUO-induced Scr group, which was greater than that in the NC and TFEB groups (Figure 2). Thus, these results show that TFEB decoy ODNs affect renal interstitial injury and fibrosis in a UUO mouse model.

### 2.2. TFEB Synthetic ODN Attenuates UUO-Induced Tubular Injury in Kidney

To investigate the effects of TFEB synthetic ODN on UUO-induced tubular injury, IHC staining was performed to observe the expression of the biomarkers of tubular injury, NGAL, and Kim-1. Figure 1C shows that UUO surgery resulted in significantly increased NGAL deposition in the distal tubules, which was suppressed by TFEB ODN administration. In addition, Kim-1 expression in renal tissues was evaluated using IHC staining. As shown in Figure 1D, Kim-1 expression was markedly increased in the UUO and UUO + Scr groups but inhibited in the UUO + TFEB group. Additionally, we measured NGAL expression using Western blot analysis, which revealed that UUO surgery with Scr ODN administration up-regulated NGAL levels (Figure 2). These findings suggest that TFEB ODN mitigated UUO-induced kidney damage in mice.

### 2.3. TFEB ODN Attenuates Inflammation in a UUO-induced Mouse Model

To investigate the effects of TFEB ODN on the expression of inflammatory cytokines, we examined the levels of inflammatory cytokines during kidney fibrosis using Western blot analysis. UUO surgery increased interleukin (IL)-1β and TNF-α levels in UUO and UUO + Scr mice (Figure 2). In contrast, the TFEB ODN treatment significantly inhibited the secretion of IL-1β and TNF-α. These results show that TFEB ODN markedly inhibited the secretion of inflammatory cytokines.

### 2.4. TFEB Decoy ODN Decreases Autophagic Activity in UUO-Induced Renal Injury

We performed immunohistochemical staining and Western blot analysis to investigate the regulatory effects of TFEB transcription factors on UUO-induced renal fibrosis. As shown in Figure 3A, B, the expression of hypoxia-inducible factor-1α (HIF-1α), Beclin-1, and LC3 were significantly increased in UUO-induced renal fibrosis. In addition, in the UUO group, conversion of the cytoplasmic form of LC3 (LC3-I) to the pre-autophagosomal and autophagosomal membrane-bound form of LC3 (LC3-II) increased, indicating an increase in autophagic activity. In contrast, HIF-1α, Beclin1, and LC3 expression was reduced, and the conversion of LC3-I to LC3-II was increased in the UUO + TFEB group. These results indicate that TFEB suppression can modulate autophagic activity in UUO-induced renal fibrosis, thereby suggesting that TFEB decoy ODN administration suppresses autophagy which is induced by UUO-induced renal injury, resulting in decreased renal fibrosis.

To assess the molecular mechanism of TFEB decoy ODN in UUO-induced renal injury, we examined the expression of TFEB using immunofluorescence assays (Figure 4A). UUO with Scr ODN administration increased the expression of TFEB (green) in renal tissues, whereas TFEB ODN treatment suppressed it. TFEB was expressed in the cytosol and increased in the nucleus in the UUO + Scr ODN group. In contrast, TFEB expression decreased in the UUO + TFEB decoy ODN mice. We also measured TFEB expression using Western blot analysis, which revealed that UUO + Scr ODN upregulated TFEB expression, whereas TFEB decoy ODN treatment decreased it during UUO-induced renal injury (Figure 4B, C). These results suggest that synthetic TFEB decoy ODN may protect the kidneys during UUO through suppression of the TFEB signaling pathway.

## 3. Discussion

Many cellular and molecular events occur during renal fibrosis, such as the activation of interstitial myofibroblasts, EMT and/or endothelial-mesenchymal transition, ECM deposition and microvascular dysfunction [52,53]. Notably, autophagy can also cause renal fibrosis after injury [54]. Dysregulation or failure of the autophagy pathway or mutations in autophagy-related genes result in various human pathologies, including cancer, neurodegenerative diseases, chronic inflammatory diseases, and cardiac failure [55,56,57].

Autophagy has renal protective effects on renal tubular cells during AKI [58], and helps repair and regenerate cells and tissues [59]. Therefore, impaired autophagy in the kidneys results in inflammation and TIF in CKD models [59]. Xu et al. [60] reported that defects in autophagy can lead to excessive deposition of ECM and renal fibrosis. In contrast, some studies have indicated that autophagy is involved in promoting fibrosis [61,62,63]. The actual function of autophagy may depend on the specific type and stage of the fibrotic disease. It has also been established that autophagy in tubules protects against AKI and cell death [64,65], and pharmacological inhibition of autophagy is used to understand the role of autophagy in kidney IRI injury [66]. The cisplatin-treated AKI model revealed that the inhibition of autophagy enhanced kidney injury remarkably, whereas the activation of autophagy protected proximal tubules from injury [67]. Wang et al. [68] demonstrated that TFEB promoted autophagy and attenuated IRI by reducing inflammation and kidney injury. In our study, we suppressed the function of TFEB using synthetic decoy ODN to evaluate autophagic activity in the process of renal injury, which prevented renal fibrosis.

Autophagy serves a dual purpose: it may play a cytoprotective role in the body [69] or promote cell injury and the development of CKD [70]. The role of autophagy in TIF is complex and inconsistent. A previous study [61] reported that autophagy in the proximal tubules may promote fibrosis by coordinately activating tubular cell death, interstitial inflammation, and, in particular, the production of pro-fibrotic factors such as fibronectin. Kim et al. [70] indicated that renal fibrosis is accompanied by the up-regulation of autophagy, whereas another study suggested that the downregulation of autophagy occurs in diabetic nephropathy [71]. These differences may be associated with the duration and stage of renal disease; autophagic down-regulation is mainly observed in the early stages of diabetes, and enhanced autophagy is often observed in the late stages of diabetes and is associated with diabetic kidney fibrosis [71]. In our study, TFEB down-regulation resulted in decreased autophagy and a subsequent decrease in renal fibrosis. Therefore, our results suggest that autophagy is closely related to renal fibrosis. However, further studies are needed to better understand its pathogenesis.

To determine the role of autophagy in renal fibrosis, most studies have used the UUO model [72], which exhibits time-dependent induction of autophagy accompanied by leukocytic infiltration, tubular cell death, tubular atrophy, and TIF [73,74]. Peng et al. [75] found that autophagy deficiency in proximal tubular epithelial cells resulted in dramatically increased leukocyte infiltration and proinflammatory cytokines expression in UUO kidneys. We also noted increased cytokine expression in our UUO mouse model. However, some studies have shown results that contradict our findings. Lu et al. [76] suggested that TFEB is a protective transcription factor against endothelial cell inflammation. A possibility of passive consequence of reduced fibrosis could be considered for the anti-inflammatory effect of TFEB inhibition, because less fibrosis and debris reduces macrophage activation. Further studies are mandatory.

Transcription factors are nuclear proteins that play an important role in the regulation of gene transcription. Synthetic decoy ODNs are repressors of transcription that bind to transcription factors and inhibit gene expression by occupying the DNA-binding site of the transcription factor in the nucleus [77]. In this study, the expression of TFEB transcription factors was suppressed using synthetic TFEB decoy ODNs injected into the tail vein of UUO mice, and the inhibition of autophagy in UUO-induced renal fibrosis was confirmed.

A previous study [78] demonstrated an elevation in activated autophagy biomarkers, such as LC3, Atg3, Atg5, Atg7, Atg12, and Atg16, in the renal tissues of UUO mice, suggesting that autophagic activation may be associated with renal tissue injury and fibrosis. LC3 and Beclin-1 are important and dependable markers of autophagy [79]. LC3 is the most widely used autophagic marker, and its conversion from the cytosolic form (LC3-I) to the lipidated form (LC3-II) is a marker of autophagosome formation [80]. Beclin1 is essential for the recruitment of other autophagy-related proteins that play a role in the expansion of autophagosomal membranes and structures [81]. In our study, LC3 and Beclin-1 were used as biological markers of autophagy. We used immunological detection methods to evaluate chemical mediators, such as HIF-1α, TNF-α, IL-1β, NGAL, Kim-1, collagen I, and fibronectin, to observe changes in UUO-induced renal injury. We observed a decrease in the expression of the autophagic markers, LC3-I/II and beclin-1, following TFEB suppression with decoy ODN injection. These results suggest that TFEB could be a promising therapeutic target for preventing renal fibrosis following injury.

TFEB plays a pivotal role in regulating the process of autophagy [39,41] by binding to the promoter regions of numerous autophagy genes and inducing autophagosome biogenesis and autophagosome–lysosome fusion [41]. Under normal conditions, TFEB is localized in the cytoplasm in its inactivated form. Under conditions of starvation or oxidative stimulation, TFEB is dephosphorylated, translocates to the nucleus, and promotes the transcription of genes related to autophagy and lysosome biogenesis [82]. In our study, TFEB was increased in UUO kidneys and translocated to the nucleus. These findings suggest that the injurious role of TFEB in UUO-induced renal damage is associated with excessive autophagy, consequent cell death and inflammation.

In summary, we designed a novel synthetic noncoding RNA targeting TFEB, a transcription factor known to induce fibrosis and inflammatory responses. We confirmed that TFEB decoy ODN administration in a UUO mouse model reduced fibrosis and autophagy (Figure 5). The TFEB decoy ODNs also showed anti-inflammatory effects. Collectively, our results suggest that TFEB may be involved in the regulation of autophagy and fibrosis and that regulating TFEB activity may be a promising therapeutic strategy against kidney diseases. However, further studies are needed to better understand the function and mechanism of TFEB in the treatment of CKD.

## 4. Materials and Methods

### 4.1. Synthesis of Ring-type TFEB Decoy ODNs and Scrambled ODNs

Decoy ODNs were designed in our laboratory and synthesized by Macrogen Co. Ltd. (Seoul, Korea). We designed a hair-pin-shaped ring-type structure TFEB decoy ODN and synthesized a double-stranded decoy ODN that contained a sequence of the TFEB-binding elements (Figure 6). The consensus sequences of the TFEB decoy ODNs and scrambled (Scr) ODNs used in this study are detailed in Table 1 (the target sites of consensus sequences are bolded and underlined). After denaturing at 95 °C for 3 min, the ODNs were annealed for 6 h, and the temperature was gradually reduced from 85 °C to 25 °C. Following the addition of T4 ligase (1U; Takara Inc., Kusatsu, Japan), the ODNs were incubated at 16 °C, for 16 h, to generate covalently ligated ring-type decoy molecules.

### 4.2. Animal Model

Male C57BL/6N mice (6 weeks old, 20–22 g; Samtako, Korea) were housed individually in humid cages and maintained at a set temperature, with a 12 h light–dark cycle. One week after acclimatization, a total of 35 mice were randomly divided into five groups, with 5 mice per group. The first group was the normal control (NC) group, and the second group was injected with TFEB decoy ODNs (TFEB group). The third group underwent UUO surgery (UUO group). The fourth group included mice that underwent UUO surgery and were injected with Scr ODNs (Scr group). The fifth group comprised mice that underwent UUO surgery and were injected with TFEB decoy ODNs (UUO + TFEB group).

For the UUO surgery, each mouse was anesthetized, its flank was incised, and the left ureter was isolated and ligated with a 5-0 silk suture at both proximal and distal locations. The TFEB ODNs (10 μg/μL) and Scr ODNs (10 μg/μL) were injected three times into the tail veins 2 days before ureteral ligation, and 2 and 5 days after UUO. The ODNs were transferred via tail vein injection, using an in vivo gene-delivery system (Mirus Bio, Madison, WI, USA). Seven days after UUO, both kidneys of each mouse were harvested and prepared for the study as described below. The animal protocols were approved by the Institutional Review Board of the Catholic University of Daegu, Korea (EXP-IRB number and protocol code: DCIAFCR-210503-02-Y, approval date: 3 May 2021).

### 4.3. Histological Analysis

All harvested kidney specimens were fixed in a 10% formalin solution for 24 h at room temperature. The fixed tissues were dehydrated with ethanol, removed with xylene, and embedded in paraffin. Paraffin-embedded tissues were cut into 4 μm sections and the sections were stained with hematoxylin and eosin (H&E) and Masson’s trichrome stains according to standard protocols. All the slides were inspected with the scanned images using a Pannoramic^®^ MIDI slide scanner (3DHISTECH Ltd., Budapest, Hungary).

### 4.4. Immunofluorescence Staining

The paraffin-embedded kidney tissue sections were placed in a blocking serum (5% bovine serum albumin (BSA) in phosphate-buffered saline) for 1 h at room temperature. Tissue sections were incubated with anti-TFEB (1:200 dilution; H00007942-M01, Novus Biologicals, Littleton, CO, USA) for 2 h at room temperature. Goat anti-mouse secondary antibody was conjugated to Alexa Fluor 488 (Invitrogen, Waltham, MA, USA). Tissue sections were stained with the nucleic acid stain Hoechst 33342. The slides were mounted using a mounting medium (DAKO, Agilent, Santa Clara, CA, USA). The stained slides were examined under a confocal fluorescence microscope (Nikon, Tokyo, Japan).

### 4.5. Immunohistochemical (IHC) Staining

Paraffin-embedded tissue sections of 4-μm thicknesses were deparaffinized with xylene, dehydrated to gradually decreasing concentrations of ethanol, and incubated with 3% hydrogen peroxidase in methanol for 10 min to block endogenous peroxidase activity. The tissue sections were immersed in a 10 mM sodium citrate buffer (pH 6.0) for 5 min at 95 °C. The last step was repeated with a 10 mM sodium citrate solution (pH 6.0). The sections were kept in the same solution while cooling for 20 min, after which they were rinsed in PBS. The sections were then treated with a primary antibody (1:100 dilution) for 1 h at 37 °C. The following primary antibodies were used: anti-neutrophil gelatinase-associated lipocalin (NGAL, 1:500 dilution; Santa Cruz Biotechnology, Dallas, TX, USA), anti-kidney injury molecule-1 (Kim-1, formerly called Tim-1, 1:3000 dilution; Abcam, Cambridge, UK), microtubule-associated protein light chain 3 (LC3)A/B (1:300 dilution; Cell Signaling Technology, Inc., Danvers, MA, USA), and Beclin1 (1:600 dilution; Cell Signaling Technology). The signal was visualized using an Envision System (DAKO, Agilent) for 30 min at 37 °C. 3,3′-Diaminobenzidine tetrahydrochloride (DAB) was used as the coloring reagent, and hematoxylin was used as the counter-stain. The slides were examined using a slide scanner (Pannoramic MIDI) and analyzed using iSolution DT software (IMTechnology, Vancouver, BC, Canada).

### 4.6. Western Blot Analysis

Kidney protein samples were extracted using lysis buffer (CelLytic™ MT, Sigma-Aldrich, St Louis, MI, USA) and centrifuged at 13,000 rpm, at 4 °C, for 10 min after incubation on ice for 30 min. The supernatant was collected and the protein concentration was measured using a Bio-Rad Bradford kit (Bio-Rad Laboratories, Hercules, CA, USA) at 595 nm using a spectrophotometer. The samples were boiled for 10 min, and equal volumes were loaded on precast gradient polyacrylamide gels (Bolt™ 4-12% Bis-Tris Plus Gels; Thermo Fisher Scientific, Waltham, MA, USA) before being transferred to a nitrocellulose membrane (GE Healthcare, Chicago, IL, USA). The membrane was blocked for 2 h at room temperature in 5% BSA and incubated with primary antibodies (1:1000) overnight, at 4 °C. Horseradish peroxidase-conjugated secondary antibodies (1:1000 dilution) were used in this study. The signal intensity was detected using an image analyzer (ChemiDoc™XRS+, Bio-Rad Laboratories) and quantified using the Image Lab software (Bio-Rad Laboratories). The primary antibodies used in this study were: anti-LC3B, anti-Beclin1, and anti-α tubulin (Cell Signaling Technology, Inc.); anti-fibronectin, anti-TNF-α, anti-Kim-1, and anti-collagen I (Abcam); anti-IL-1β, anti-HIF-1α, and anti-NGAL (Santa Cruz Biotechnology lnc., USA); anti-TFEB (Novus Bio, Littleton, CO, USA).

### 4.7. Statistical Analysis

All data are presented as mean ± standard error. Statistical significance was determined via one-way ANOVA with Turkey’s multiple comparison test using GraphPad Prism 5.0 (GraphPad Software, Inc., San Diego, CA, USA). Turkey’s multiple tests were conducted only when F achieved *p* < 0.05 and when there was no significant variance inhomogeneity. Statistical significance was set as *p* < 0.05.

## Figures and Tables

**Figure 1 ijms-23-08138-f001:**
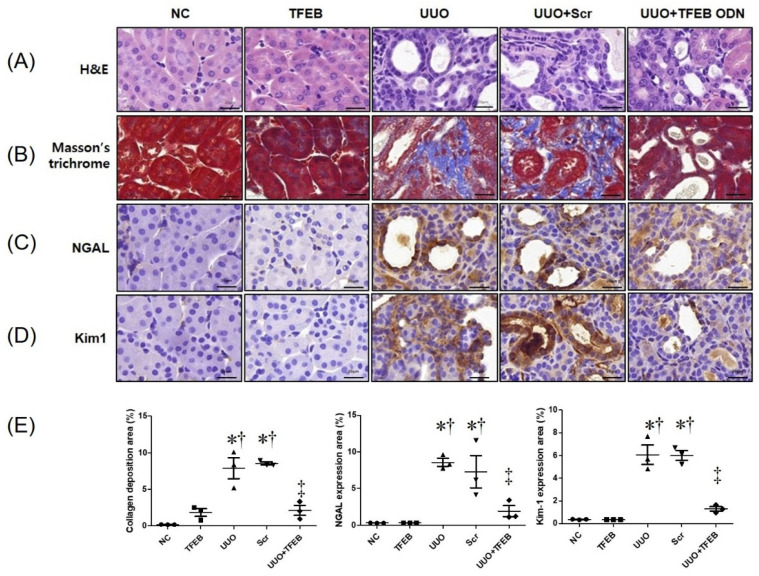
TFEB decoy ODN attenuated morphological changes and tubular injury in UUO-induced renal fibrosis in mice. (**A**) Paraffin-embedded kidney section stained with hematoxylin and eosin (H&E); (**B**) Masson’s trichrome staining; immunohistochemical staining showing the protein expression of (**C**) NGAL and (**D**) Kim-1 protein; (**E**) quantification of collagen, NGAL, and Kim-1 expressions. Quantitative analysis of collagen or protein expression area of each group (*n* = 3; 400× magnification). Original scale bar = 20 μm. The results are expressed as mean ± SE of three independent experiments. * *p* < 0.05 vs. the NC group. † *p* < 0.05 vs. the TFEB group. ‡ *p* < 0.05 vs. the UUO or Scr group. NC, normal control group; TFEB, TFEB decoy injection in the normal control group; UUO, UUO surgery group; Scr, scramble ODN injection in the UUO surgery group; UUO + TFEB, TFEB decoy ODN injection in the UUO surgery group; UUO, unilateral ureteral obstruction; decoy ODNs, decoy oligonucleotides; TFEB, Transcription factor EB; Scr, scrambled.

**Figure 2 ijms-23-08138-f002:**
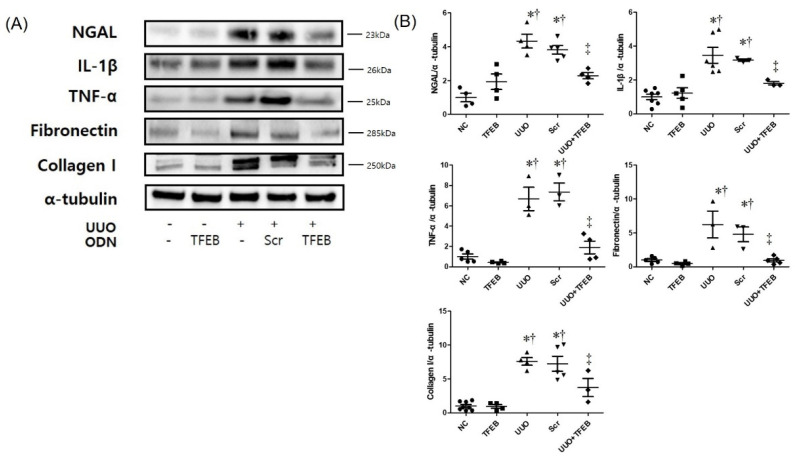
Synthetic TFEB decoy ODN significantly suppressed renal inflammation, fibrosis, and collagen deposition in UUO mice. (**A**) Western blotting showing NGAL, inflammatory cytokines (IL-1β, TNF-α), fibronectin, and collagen I expression in kidney tissue; (**B**) quantification of the protein expression levels of the Western blot images using Image J program. The results are expressed as mean ± SE of three independent experiments. * *p* < 0.05 vs. the NC group. † *p* < 0.05 vs. the TFEB group. ‡ *p* < 0.05 vs. the UUO or Scr group. NC, normal control group; TFEB, TFEB decoy injection in the normal control group; UUO, UUO surgery group; Scr, scramble ODN injection in the UUO surgery group; UUO + TFEB, TFEB decoy ODN injection in the UUO surgery group; UUO, unilateral ureteral obstruction; decoy ODNs, decoy oligonucleotides; TFEB, Transcription factor EB; Scr, scrambled; IL-1β, interleukin-1β; tumor necrosis factor-α (TNF-α).

**Figure 3 ijms-23-08138-f003:**
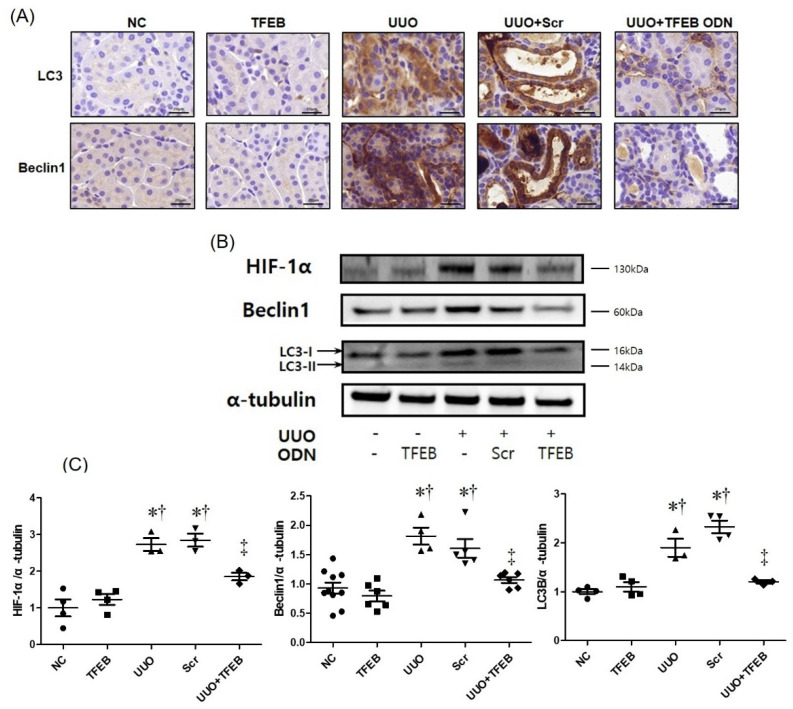
TFEB decoy ODNs decreased the expression of autophagic markers in UUO-induced renal injury. TFEB decoy ODN inhibits the expression of autophagy genes in UUO-induced renal fibrosis. (**A**) Immunohistochemical staining shows that the expressions of LC3 and Beclin1 proteins were reduced via TFEB decoy ODN administration in UUO-induced renal fibrosis mice. Original scale bar = 20 μm. (**B**) Western blot analysis shows that TFEB decoy ODN decreased the expression of HIF-1α, Beclin1, and LC3. (**C**) Quantification of the protein expression levels of the Western blot images using the Image J program. The results are expressed as mean ± SE of three independent experiments. * *p* < 0.05 vs. the NC group. † *p* < 0.05 vs. the TFEB group. ‡ *p* < 0.05 vs. the UUO or Scr group. NC, normal control group; TFEB, TFEB decoy injection in the normal control group; UUO, UUO surgery group; Scr, scramble ODN injection in the UUO surgery group; UUO + TFEB, TFEB decoy ODN injection in the UUO surgery group; LC3, microtubule-associated protein light chain 3; HIF-1α, hypoxia-inducible factor-1α; UUO, unilateral ureteral obstruction; decoy ODNs, decoy oligonucleotides; TFEB, Transcription factor EB; Scr, scrambled.

**Figure 4 ijms-23-08138-f004:**
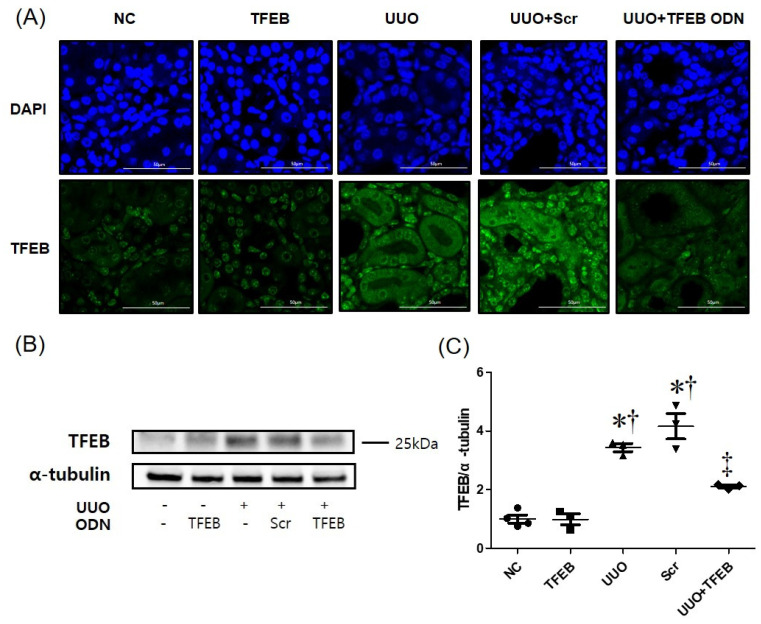
The effect of TFEB ODN on TFEB expression in UUO mice. (**A**) Immunofluorescence staining for TFEB (green). Cells were counterstained with Hoechst 33342 (blue). Original scale bar = 20 μm. (**B**) Western blot analysis shows that TFEB decoy ODN decreased TFEB expression. **(C)** Quantitative analysis of TFEB expression in each group, performed at a magnification of 400×. Data are presented as mean ± SEM (*n* = 3). Tukey’s multiple comparison test, * *p* < 0.05 vs. the NC group. † *p* < 0.05 vs. the TFEB group. ‡ *p* < 0.05 vs. the UUO or Scr group. NC, normal control group; TFEB, TFEB decoy injection in the normal control group; UUO, UUO surgery group; Scr, scramble ODN injection in the UUO surgery group; UUO + TFEB, TFEB decoy ODN injection in the UUO surgery group; DAPI, 40, 6-diamidino-2-phenylindole; UUO, unilateral ureteral obstruction; decoy ODNs, decoy oligonucleotides; TFEB, Transcription factor EB; Scr, scrambled.

**Figure 5 ijms-23-08138-f005:**
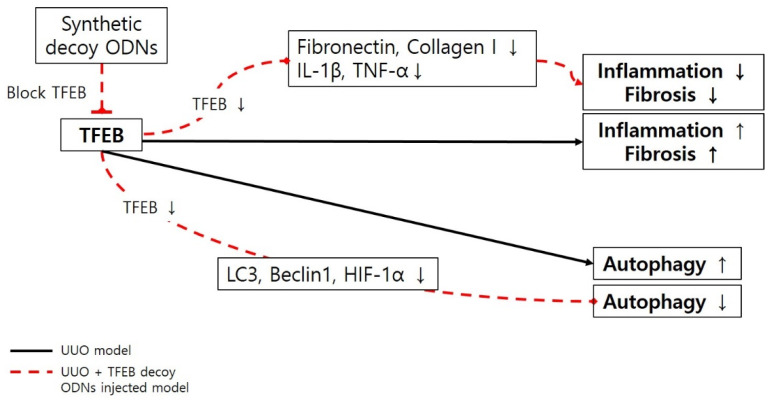
Schematic diagram of the molecular pathway for TFEB transcription factor and inhibitory effects of synthetic decoy ODNs on autophagy in renal injury UUO, unilateral ureteral obstruction; decoy ODNs, decoy oligonucleotides; TFEB, Transcription factor EB; IL-1β, interleukin-1β; TNF-α, tumor necrosis factor-α; LC3, microtubule-associated protein light chain 3; HIF-1α, hypoxia inducible factor-1α.

**Figure 6 ijms-23-08138-f006:**
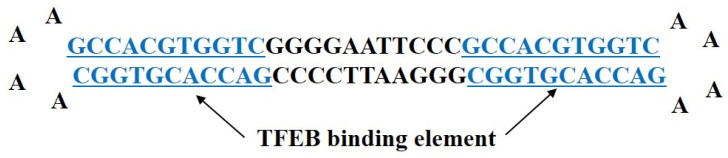
Synthesis of TFEB decoy oligodeoxynucleotide (ODN). The design of ring-type TFEB decoy ODNs including GCCACGTGGTC (underlined), which is the consensus sequence for the TFEB-binding element.

**Table 1 ijms-23-08138-t001:** Sequences of the ODNs used in this study. The target site of the synthesized TFEB decoy ODN sequence is underlined.

Decoy	Sequence
Scr	5′-GAATTCAATTCAGGGTACGGCAAAAAATTGCCGTACCCTGAATT-3′
TFEB	5′-GAATTCCC***GCCACGTGGTC***AAAA***GACCACGTGGC***GGGAA TTCCCC***GACCACGTGGC*** AAAA***GCCACGTGGTC***GGG-3′

TFEB, transcription factor EB; Scr, scrambled; ODNs, oligonucleotides.

## Data Availability

Not applicable.

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
