# Peer review of "Anti-Fibrotic Effect of Synthetic Noncoding Decoy ODNs for TFEB in an Animal Model of Chronic Kidney Disease"

_ijms, 2022, doi:10.3390/ijms23158138_

Round 1
Reviewer 1 Report
The maunscript is well written and easy to follow. I have some minor concerns and comments:
1) The animal strain could be either C57Bl6/J or C57Bl6/N (one is fibrosis resistant) so this should be clarified.
2) The chosen 7 day post-UUO time-point for kidney analysis already shows remarkable fibrosis, an additional earlier analysis would enhance the paper (eg. 1 day or 3 days post-UUO).
3) Please describe how was the ODN administered, what carrier molecule was used (eg Invivofectamine?)
4) For TFEB immunohisto, using primary anti-mouse antibody on mouse tissue needs a careful secondary antibody choice due to strong background. Please provide clone and catalogue numbers for each antibody.
5) The used statistical anaylsis is inappropriate for non-Gaussian distrubuted data such as expressions or histology analyses. Please recalculate statistics using non-parametric Kruskal-Wallis test!
6) For data presentation, please change bar charts to scatter plots. Given the high SEM values in some graphs, concerns rise about significance so one must see the distribution of individual data. Please correct!
7) Please provide molecular weights for each immunoblot. Also on the original blot file.
8) How would you explain that TFEB suppression reduced renal inflammation? TFEB was shown to reduce inflammatory response in endothelial cells (Lu et al. Science Signaling 2017). Is it possible that less fibrosis and debris reduces macrophage activation, so reduced inflammation is only a passive consequence of reduced fibrosis? Please discuss.
9) A proposed pathomechanism summary figure would improve discussion as well.
10) The last summary paragraphs defines ODN as "novel synthetic noncoding RNA" although it looks more a DNA sequence to me. Please clarify!
Reviewer 2 Report
The authors have analyzed the protective role of synthetic noncoding decoy ODNs for
TFEB in the mouse model of UUO.
1. Major Strength: Discussed well, add useful new information
2. Limitations: The introduction section and mechanisms are poorly addressed.
Comments
3. Fig 2; Analyze the effect of TFEB ODN on the contralateral kidney. Include Sirius red staining, and the levels of fibrosis markers and vimentin levels
4. Fig 2: analyze collagen levels in these groups, and FSP-1-E-cadherin co-labelling or alpha Sma-E-cadherin IF staining in key experiments to show EMT.
5. Analyze endothelial-to-mesenchymal transitions in the key experiments; alpha SMA-CD31 IF staining
6. Include ACR levels, plasma creatine levels, and glomerular functions in your key experiment.
7. Measure the IL-6, IL-1b, and TNFa levels in the plasma and kidney lysate. Western blot is not the correct strategy to analyze secretory proteins.
8. Also Analyze these above mention experiments and assays in the contralateral kidneys. Such as EMT and EndMT analysis.
9. Role of TFEB and TFEB CDNs against EMT and EndMT.
10. IL-6 is an important cytokine, involved in the defects in autophagy levels and fatty acid oxidation. Glucocorticoid receptor (GR) signaling is important in regulating IL-6 levels. Analyze GR levels and FAO levels in your key samples.
11. Your data suggests that TFEB negatively interacts with the HIFa. HIF1a is key glycolytic molecule. I would suggest analyzing the glycolysis levels in your key experiments.
12. Defective glycolysis and FAO levels are one of the critical phenomena causing myofibroblasts formations. SIRT3 protein regulates it. Analyze the level of SIRT3 in the key samples.
Take your time and improve the mechanism part, which is poor in the current manuscript.
Introduction: elaborate this section
Include one para about therapeutics. Elaborate briefly following Potential drugs that have been evaluated to be included such as a) SIRT3 and glycolysis inhibitors, c) Linagliptin d) ROCK inhibitors, e) mineralocorticoid antagonists, and f) peptide AcSDKP in protecting against kidney injuries.
1) Comparative efficacy of ACE inhibitors vs ARBs.
2) Mechanisms such as EMT and EndMT in renal fibrosis
3) In few lines, briefly describe the pathways in DN, such as a) TGFB and WNT signaling, b) endothelial Glucocorticoid receptor c) podocyte glucocorticoid receptor c) FGFR1 signaling, d) Notch signaling and Hedgehog signaling f) Endothelial SIRT3 mediated mechanisms and DPP-4 mediated mechanism
4) Include the rationale of EMT and EndMT in your models. GR, SIRT3, and FGFR1 are negative regulators whereas DPP-4, and integrin B1 activate these processes.
5) Write the background rationale of profibrotic TNFa, IL-1B, and IL-6 in kidney injuries.
IL-6 neutralization in mice reverses the fibrotic phenotype in the kidney.
Round 2
Reviewer 2 Report
The authors should be careful during revising.
The ethical statement for animals is missing. Include it with the protocol number.
Antibody dilution has not been given
Authors should be careful citing their works. The authors have written several mechanisms and molecules but did not place the respective original works as citations. In the introduction section: line no- 40-43-Cite the correct respective original works for TGF B signaling and Wnt signaling in dkd; endothelial Glucocorticoid receptors; podocyte glucocorticoid receptor; endothelial SIRT3; DPP-4 and endmt in diabetic kidney
Similarly lines 59-62- the authors mentioned several inhibitors as a treatment option for DKD. Include the original works, not reviews. Include the citations wherever missing.
Check the English again. In some places, English is not clear.
